# Overcoming Recalcitrance: A Review of Regeneration Methods and Challenges in Roses

**DOI:** 10.3390/plants14243797

**Published:** 2025-12-13

**Authors:** Anna Nelson, Thomas Ranney, Wusheng Liu, Tim Kelliher, Hui Duan, Kedong Da

**Affiliations:** 1Plant Transformation Lab, North Carolina State University, Raleigh, NC 27695, USA; ainelson@ncsu.edu; 2Department of Horticultural Sciences, North Carolina State University, Raleigh, NC 27695, USA; tranney@ncsu.edu (T.R.); wliu25@ncsu.edu (W.L.); tjkellih@ncsu.edu (T.K.); 3Genome Editing Center, North Carolina State University, Raleigh, NC 27607, USA; 4USDA-ARS, U.S. National Arboretum, Beltsville, MD 20705, USA; hui.duan@usda.gov

**Keywords:** *Rosa*, *in vitro* culture, regeneration, cultivar, explants

## Abstract

Roses (*Rosa* spp.) are among the most economically and ornamentally important floricultural crops worldwide, yet their improvement is constrained by inefficient breeding methods. Tissue culture regeneration based plant transformation and genome editing technologies provide innovative and increasingly effective approaches to surmount these longstanding challenges; however, rose tissue culture regeneration remains notoriously recalcitrant. Successful plant regeneration in roses depends on multiple factors, including genotype, explant source, physiological status, and the precise combination of plant growth regulators and culture conditions. Over the past three decades, numerous efforts have focused on optimizing rose organogenesis and somatic embryogenesis systems. Despite progress, low regeneration frequencies, strong genotype dependency continue to limit molecular breeding and genome editing application in rose. This review synthesizes current advances in *in vitro* regeneration methods for roses, emphasizing key determinants of morphogenic response, including explant selection, hormonal balance, media composition, light and temperature regimes, and the organic and inorganic additives. The underlying causes of recalcitrance are discussed in relation to tissue physiology, biochemical and molecular regulation of morphogenesis. Finally, strategies for overcoming regeneration barriers—such as the use of morphogenic regulators and *in planta* transformation—are highlighted as emerging avenues toward cultivar independent transformation and genome editing for rose.

## 1. Introduction

Roses are cultivated extensively for their ornamental, pharmaceutical, and industrial applications. Ornamentally, their commercial value is attributed to their attractive flowers, extended bloom periods, and fragrance [1,2]. Roses also contain a variety of bioactive compounds with demonstrated pharmacological efficacy, including anti-inflammatory and antioxidant properties [3]. Rose oil is widely utilized in perfumery, cosmetics, and therapeutic formulations [4,5]. The continued development of elite cultivars exhibiting enhanced disease resistance, optimized secondary metabolite production, improved abiotic stress tolerance, and other commercial traits is critical for meeting the demands of both horticultural and pharmaceutical markets [6]. The advancement of novel rose cultivars with traditional methods is hindered by factors such as reproductive barriers [7,8], extended generation intervals [9], and polyploidy [10,11,12]. In contrast, genetic transformation and genome editing enable the introduction of desired traits into elite varieties in a short period of time.

Almost all of the current plant transformation and genome editing methods require regeneration from tissue culture [13,14]. Tissue culture has facilitated the transformation of rose varieties [15,16]. However, the process is often limited to one or two genotypes, and those transformable genotypes are usually characterized by poor agronomic performance. The genome editing (GE) tool (defined as clustered regularly interspaced short palindromic repeats (CRISPR)/CRISPR-related nuclease 9 (Cas9)) uses optimized tissue culture systems that can only be applied to a reported transformable rose genotype [17]. The success of genetic transformation relies on the delivery of DNA into the cell and the regeneration of transgenic plants [13,14].

Rose regeneration primarily occurs through two pathways: organogenesis and somatic embryogenesis [18,19]. Although regeneration protocols have been established for several varieties, including *R.* ‘Samantha’, *R. chinensis* ‘Old Blush’, *R. hybrida* ‘Eiffel Tower’, ‘Carefree Beauty’, ‘Italian Ice’, and ‘Ringo All-Star’ [11,20,21], most systems still exhibit low regeneration frequencies and remain recalcitrant to genetic transformation, underscoring the need to develop and optimize more efficient regeneration methods with broader varietal applicability.

## 2. Limiting Factors in Rose Regeneration

The most challenging limitation in *Rosa* regeneration protocols is the strong genotype dependency of regenerative competence. This is a direct consequence of the complex domestication history. Modern rose cultivars are rarely genetically similar; they are highly heterozygous, often polyploid hybrids resulting from centuries of introgression between disparate species (e.g., *R. chinensis*, *R. multiflora*, *R. moschata*, and *R. foetida*) [11]. This genetic mosaicism leads to highly unpredictable *in vitro* responses, even within the same horticultural class. Regeneration remains highly recalcitrant for many genotypes, as the hormonal ratios required to trigger the necessary transcriptomic reprogramming (such as *WIND1* or *WUS* activation) vary wildly depending on the specific genetic heritage and ploidy level of the donor plant [22,23].

There is a lack of standardization in reported culture conditions, particularly regarding the photoperiodic and spectral environment. While the chemical composition of media is usually detailed, physical parameters such as light spectrum, photosynthetic photon flux density (PPFD), and specific light/dark cycles are frequently unreported or described only in vague terms [24,25]. Light acts not merely as an energy source but as a potent morphogenic signal that regulates reactive oxygen species homeostasis and hormonal sensitivity [26]. The failure to report these photomorphogenic cues prevents the replication of successful protocols and obscures the specific environmental triggers required to overcome recalcitrance in difficult cultivars.

The success of regeneration is further dictated by the physiological and ontogenetic age of the explant material. There is a marked divergence in competency between juvenile tissues such as immature zygotic embryos or seedling hypocotyls and mature, woody tissues derived from established field-grown plants [27,28]. Mature explants possess rigid epigenetic landscapes that are characterized by high levels of DNA methylation and stable repressive complexes that are resistant to the dedifferentiation signals required for organogenesis or somatic embryogenesis [29,30].

The field suffers from a scattershot approach to cultivar selection, which hinders the establishment of a cohesive biological model for rose regeneration. Research efforts are often fragmented across a disparate array of genotypes: some studies focus on high-value commercial cut flowers for their marketable utility, while others focus on heritage or garden varieties [31,32]. Other researchers target obscure lines useful only for long-term breeding schemes (e.g., rootstocks or disease-resistant wild species) [33,34]. This lack of a centralized model genotype means that molecular insights gained in one study are rarely directly applicable to others. The result is a collection of isolated success stories rather than a unified, systematic understanding of the molecular mechanisms governing regeneration across *Rosa* genotypes.

## 3. Model of Plant Regeneration in Rose Tissue Culture

Plant tissue culture regeneration relies on cultivating plant cells, tissues, or organs in a nutrient medium under aseptic conditions. Rose regeneration can be achieved through organogenesis and somatic embryogenesis [31,35]. Organogenesis is a process for regenerating organs (shoots, roots, or whole plants) from plant tissues or cells grown *in vitro*, and can be further divided into direct organogenesis, where organs form directly from the explant, and indirect organogenesis, where organs form through a callus intermediate (Figure 1). Somatic embryogenesis is a process in which somatic (non-reproductive) cells develop into embryo-like structures that can grow into complete plants, which are used for both clonal propagation and genetic transformation [36,37,38]. Two different Rose *in vitro* culture regeneration systems have been reported: solid culture and liquid culture. The choice between solid and liquid media significantly impacts the growth and differentiation of plant cells [39,40]. Solid media provides a stable support matrix, which facilitates explant organogenesis, direct regeneration, or callus formation [41]. Compared to solid cultures, liquid cultures offer increased nutrient availability and aeration [42], which promote rapid embryogenic cell proliferation and are particularly useful for protoplast manipulation and cell regeneration-based genome editing [43,44,45].

Direct regeneration eliminates the callus phase, reducing the frequency of somaclonal variation, or potential genetic changes arising from tissue culture. It is often preferred for maintaining genetic fidelity. Indirect regeneration increases somaclonal variations but may allow for high-throughput regeneration and is often necessary for genotypes recalcitrant to direct regeneration. Callus may provide a source of cells for genetic transformation and secondary metabolite production. The decision between direct and indirect regeneration depends on the specific plant species, genotype, and experimental goals. Figure 1 shows a detailed explanation of regeneration pathways and types.

## 4. Molecular Pathways of Regeneration

Differentiated tissues have developmental genes that are transcriptionally silenced by Polycomb Repressive Complexes (PRCs). Regeneration is triggered by external stimuli, either exogenous hormone application or wounding, which destabilizes the repressor proteins and activates multiple molecular mechanisms. *De novo* shoot organogenesis is a two-step process where somatic explants form pluripotent callus on a media containing hormonal regulators. Subsequent transfer to media high in cytokinin results in the activation of multiple meristematic identity genes, such as *WUSHEL* (*WUS*) and *CUP-SHAPED COTYLEDON1* (*CUC1*).

The shoot regeneration pathway was initially characterized in *Arabidopsis thaliana* (Figure 2). Local wound stress induces the upregulation of *WOUND INDUCED DEDIFFERENTIATION1* (*WIND1*). WIND1 subsequently activates *ENHANCER OF SHOOT REGENERATION1* (*ESR1*), which is further enhanced by the auxin- and cytokinin-rich environment of the callus induction medium (CIM). Cytokinin also activates the B-type response regulators *ARABIDOPSIS RESPONSE REGULATOR1* (*ARR1*), *ARR10*, and *ARR12*, which further amplify *ESR1* expression. Auxin signaling activates multiple *AUXIN RESPONSE FACTORS* (*ARFs*), which in turn trigger the expression of a suite of *LATERAL ORGAN BOUNDARIES DOMAIN* (*LBD*) genes, including *LBD16*, *LBD17*, *LBD18*, and *LBD29*. The activation of these pathways results in the formation of a pluripotent callus. This callus is then transferred to a shoot induction medium (SIM), which possesses a high cytokinin-to-auxin ratio. On shoot induction medium, *ESR1* expression is maintained and further upregulates the shoot meristem identity genes *ESR2*, *CUC1*, *WUS*, *SHOOT MERISTEMLESS* (*STM*), and *RELATED to APETALA2.6-LIKE* (*RAP2.6L*). These genes promote reorganization of the callus into a shoot [46]. 

Somatic embryogenesis is a non-linear pathway (Figure 3). The initiation of somatic embryogenesis is contingent upon the cellular competency of the explant tissue. Competency can be improved by either hormonal or stress stimulus. 2,4-D is a synthetic auxin commonly used to induce shifts in chromatin accessibility. These external factors allow for the expression of master transcription factors, e.g., *WUSCHEL* (*WUS)* and *BABY BOOM* (*BBM*), which establish a stable, self-reinforcing genetic network by activating the proteins like *AGAMOUS-LIKE15* (*AGL15*) and the *LAFL* genes, *LEAFY COTYLEDON1* (*LEC1*), *ABSCISIC ACID INSENSITIVE 3* (*ABI3*), *FUSCA3* (*FUS3*), and *LEC2*.

In somatic tissues, the master regulator genes are transcriptionally silenced by several repressors. The most common proteins that repress gene expression are POLYCOMB REPRESSIVE COMPLEXE1 (PRC1), PRC2, PICKLE (PKL), VIVIPAROUS/ABI3-LIKE1 (VAL1), and VAL2. An external hormonal stress or a wounding response triggers a cellular reprogramming that displaces those repressors. This induction is marked by the activation of *SOMATIC EMBRYO RECEPTOR KINASE* (*SERK*) genes and the upregulation of chromatin modifiers like *HISTONE ACETYLTRANSFERASE1* (*HAC1*) and *AT-HOOK MOTIF NUCLEAR LOCALIZED15* (*AHL15*). With *WUS* and *BBM* no longer being repressed, these regulatory genes form a positive feedback loop. WUS regulates the transition from vegetative tissue to embryogenic tissue. It transcriptionally induces *LEC1*, *LEC2*, and *AGL15*. BBM activates the *LAFL* genes. LEC1 opens up the chromatin and upregulates the function of other genes in the *LAFL* complex. ABI3 and FUS3 are involved in embryo resource accumulation and desiccation tolerance. LEC2 activates the *YUC* genes that are responsible for auxin biosynthesis and the *PIN* genes that code for auxin efflux proteins. They provide polar transport and the auxin gradient necessary for embryo formation. Similarly, AUX1, LAX1, and LAX2 are other auxin influx proteins that work with PIN proteins [47].

Current literature indicates that very little functional genomic research has been done in roses to identify the mechanisms governing regeneration. Comparative genomics is the prevailing methodology for characterizing novel genes in non-model organisms. This strategy involves identifying orthologs of genes known to regulate morphogenesis in well-characterized model species, such as *Arabidopsis thaliana*. Following identification, the function of the candidate gene is validated through functional complementation assays. This is often accomplished by introducing the *Rosa* ortholog into an *Arabidopsis* knockout mutant that lacks the functional endogenous gene. The conserved functional role is determined based on whether the expression of the *Rosa* ortholog can rescue the mutant phenotype.

Some examples of this methodology can be found in *R. canina*, where research has focused on identifying genes that can improve regeneration efficiency. Previous research has successfully induced protocorm-like bodies (PLBs) from leaf and root explants, but the specific genetic pathway controlling this process had not been characterized [48,49]. In one study, the researchers focused on the *BBM* gene pathway. These genes are strongly associated with somatic embryogenesis and overall regenerative capacity in plants. Two distinct *RcBBM* cDNA clones designated as *RcBBM1* and *RcBBM2* within *R. canina* were identified and validated using *A. thaliana* knockouts [50]. A second research team identified a homolog of the *Arabidopsis FUS3* gene from PLBs of *R. canina. FUS3* encodes a protein that regulates embryonic gene expression. Expression of *RcFUS3* in *Arabidopsis fus3* mutants rescued the mutant phenotype, indicating that the *FUS3* function is conserved in both species. Additionally, overexpression of *RcFUS3* in wild-type *Arabidopsis* caused complete male sterility and developmental retardation of the seeds and seedlings [51]. This research team also isolated a gene designated as *RcAGL15* from protocorm-like bodies of *R. canina*, which was determined to be a regulatory gene that affects morphological development, flowering, and somatic embryogenesis in *R. canina* [52].

## 5. Biochemistry of Rose Regeneration

Superoxide dismutase (SOD) and superoxide peroxidase (POD) are enzymes responsible for managing oxidative stress and regulating developmental transitions by interacting with reactive oxygen species (ROS). Enzymatic activities of superoxide dismutase and superoxide peroxidase undergo dynamic fluctuations that correspond to shifts in cellular competency. Superoxide dismutase and superoxide peroxidase levels provide a quantitative metric to distinguish between developmental pathways and assess the regenerative capacity of the sampled plant material. Elevated activity is frequently correlated with an increase in metabolic activity and oxidative shifts required in regeneration [53,54].

Relative superoxide dismutase and superoxide peroxidase levels were used to analyze regeneration patterns in *R. hybrida* ‘J. F. Kennedy’ [55]. Two distinct regeneration phenotypes were observed in the cultivar: single-shoot embryos and multi-shoot embryos. Data revealed that while both types of regenerated plants exhibited upregulated superoxide dismutase and superoxide peroxidase activity when compared to somatic embryos that did not mature, the single-shoot embryos had significantly higher superoxide dismutase and superoxide peroxidase activity than the multi-shoot embryos. ROS studies are much more common in petal drop studies in roses. Unlike the anabolic upregulation of antioxidant defenses observed during regeneration, the catabolic process of abscission is driven by the transcriptional repression of key scavenging enzymes. In *R. hybrida* ‘Tineké’, ethylene signaling triggers a significant downregulation of genes encoding superoxide dismutase (*RhSOD1*) and downstream peroxidases such as ascorbate peroxidase (*RhAPX6.1* and *RhAPX11.3*) and catalase (*RhCATA*) [56]. Exogenous ethylene treatment is a suppressor of embryogenic induction but also allows ROS accumulation and induces cell separation.

## 6. Advances in Rose Tissue Culture (1990–2025)

This section explores how the initial plant material and its preparation methods impact the subsequent development of new tissues and organs. It also examines the role of plant growth regulators in inducing callus formation, shoot development, and somatic embryo formation. The choice of explant type, such as leaf, stem, root, or meristem, influences regeneration potential. Younger, actively growing tissues often exhibit higher regeneration rates. The physiological state of the donor plant and the explant’s position on the plant can also affect its response. Photoperiod, light quality and intensity affect morphogenesis, temperature affects growth rate and metabolism.

### 6.1. Environmental Conditions–Temperature

Published protocols are inconsistent in environmental factors. Observed temperatures for regeneration studies typically ranged between 18 and 30 °C. When reported, light conditions commonly involved a 16 h photoperiod, with intensities ranging from 11.5 μmol m^−2^ s^−1^ to 230 μmol m^−2^ s^−1^. Some genotypes require a continual photoperiod for the formation of somatic embryos or callus-shoots [57]. Other genotypes may require higher intensities of light to form embryos. For example, *R. hybrida* ‘Soraya’ was reported to develop embryos that initially were developed under cool white fluorescent light (50 μmol m^−2^ s^−1^), but those embryos only further matured into plantlets upon exposure to a higher intensity (150 μmol m^−2^ s^−1^) of the same light [26]. Other cultivars may respond better to far-red light conditions [58]. Reports of rose tissue culture on solid media highlighted the fact that embryogenic callus typically develops in dark conditions, while embryo conversion typically requires exposure to white light of varying intensities. Meanwhile, adventitious shoots can develop independently of exposure to light. Specific observations highlight these trends: callus incubated in dark conditions resulted in somatic embryo formation in multiple rose cultivars, including *R. hybrida* ‘Livin’ Easy’, *R. floribunda* ‘Trumpeter’, *R. hybrida* ‘Dr. Huey’, and *R. multiflora* ‘Tineké’ [34,59]. There were a few reports of callus induction occurring under light conditions, and when it did, the callus produced tended to be non-embryogenic. This can be observed in *R. hybrida* ‘Ingrid Bergman’ and *R. hybrida* ‘Xindongfang’ [60].

The process of adventitious shoot formation is typically composed of two stages: an initial period where explants are placed in darkness to induce callus, and a subsequent step where the newly developed callus is exposed to light, which then differentiates into shoots [23]. Instances of organogenic shoot development can be seen in multiple cultivars, including *R. chinensis* var. *minima* ‘Red Sunblaze’, *R. hybrida* ‘Saltze Gold’, *R. wichurana*, and *R. chinensis* ‘Old Blush’ [24,30,61]. Some cultivars are capable of light-independent shoot development, as seen in *R. hybrida* ‘Black Baccara’, ‘Maroussia’, and ‘Amanda’ [35].

Genotype-dependency was reported where multiple cultivars had callus induced on the same media composition in the dark at 24 °C. For example, *R. hybrida* ‘Baby Love’, ‘Ingrid Bergman’, ‘Perfume Delight’, ‘Prominent’, and ‘Sunflare’ expressed no regenerative features, but ‘Tournament of Roses’ formed somatic embryos under the same conditions [31]. Another example of genotype-specific interactions was reported in *R. hybrida* ‘Carefree Beauty’ undergoing repetitive somatic embryo formation, while the cultivar ‘Grand Gala’ only produced callus [36]. Similar environmental conditions were successful at inducing embryos in *R. hybrida* ‘Vickey Brown’ and ‘Domingo’ [37].

### 6.2. Environmental Conditions–Liquid Cultures

Several studies have explored liquid culture methods for rose regeneration. One report described the successful induction of organogenic shoots using a static liquid culture [38]. In another study, *R. floribunda* ‘Trumpeter’, *R. hybrida* ‘Dr. Huey’, and *R. multiflora* ‘Tineké’ induced callus on solid media, but also incorporated a liquid washing phase to filter and select various stages of embryos [34]. In Burger et al. (1990), suspension culture was used to proliferate embryogenic material originating from immature embryos [62]. High yields of protoplasts were isolated from cell suspension cultures of *R. persica* × *xanthina* and *R. wichuraiana* under dark conditions, but the incubation settings were not described and only *R. persica* × *xanthina* was capable of producing somatic embryos [63]. Dohm et al. (2001) reported a callus suspension system in *R. hybrida* ‘Heckenzauber’, yielding somatic embryos in liquid culture at 25 °C under ambient light conditions, with the platform shaker set to 90 rpm [22]. Another liquid culture study (25 °C, in darkness) by Schum and Hofmann (2001) evaluated five genotypes and found that *R. persica* × *R. xanthina* and *R. hybrida* ‘Pariser Charme’ developed adventitious shoots and somatic embryos from callus, *R. hybrida* ‘Elina’ produced only somatic embryos from callus, while *R. multiflora* and *R. wichurana* only formed callus [64]. The most recent report of successful suspension culture in rose demonstrated that *R. hybrida* ‘Samantha’ produced somatic embryos when the callus was maintained in liquid culture at 25 °C, dark conditions, and shaking at 130 rpm [15].

### 6.3. Explant Source and Selection

Different plant tissues and organs can be used for tissue culture and regeneration in roses (Figure 4). A review of explant sources in rose regeneration studies reveals a preference for leaf-based tissues as the primary material for regeneration studies (Table 1). Experiments have frequently used various forms of leaf explants, including leaflets, whole leaves, and leaf sections [26,65]. Somatic embryos have developed from callus that originated from leaf and leaflet explants in multiple cultivars. *R. hybrida* ‘Carola’ and ‘E.H.L. Krause RI’ were capable of producing somatic embryos from callus generated from unexpanded compound leaves [66,67]. Other cultivars like *R. chinensis* ‘Old Blush’ and *R. hybrida* ‘Carola’, ‘Tournament of Roses’ and ‘4th of July’ had leaf-derived embryogenic calli [32,68,69]. Leaves with the petiole attached were capable of producing organogenic shoots from *R. hybrida* ‘Amanda’, ‘Black Baccara’, and ‘Maroussia’ [35]. Another study determined that organogenic shoots could arise from leaf sections in *R. chinensis* var. minima ‘Baby Katie’ and ‘Red Sunblaze’ [61]. *R. hybrida* ‘Eiffel Tower’ was also capable of producing organogenic shoots from leaf disks [21].

Explants such as immature embryos, stem internodes, and roots have also been reported as sources for successful regeneration systems, although these were less commonly used due to the difficulties in excision and induction under *in vitro* conditions, lower regenerative capacity, and potential biases in the literature (Table 1). Internodal stem sections supported regeneration in *R. hybrida* ‘Landora’ [70]. Embryo- and reproductive-related tissues, including immature embryos, filaments, and petals, have been used to induce embryogenic callus, such as somatic embryos obtained from callus initiated from immature embryos of *R. bourboniana* [33]. Roots were rarely reported as explant sources, but *R. hybrida* ‘Fresham’ formed somatic embryos from root-derived embryogenic callus [71].

**Table 1 plants-14-03797-t001:** Published results related to explant and cultivar selection on rose tissue culture regeneration from 1990 to 2025.

Explant Source	Regeneration Type	Cultivar	Associated Publications
Anthers	Organogenesis; Somatic Embryogenesis	*R. hybrida* ‘Meirutral’	Arene et al., 1993 [72]
Embryos	Organogenesis	*R. hybrida* ‘Bridal Pink’ × pollen parents (also *R. hybrida*)	Burger et al., 1990 [62]
		*R. hybrida* ‘Shortcake’	Asano and Tanimoto, 2002 [73]
	Organogenesis; Somatic Embryogenesis	*R. hybrida* ‘Meirutral’	Arene et al., 1993 [72]
	Somatic Embryogenesis	*R. rugosa* Thunb.	Kunitake et al., 1993 [57]
		*R.* *rugosa*	Kim et al., 2004 [32]
		*R.* *bourboniana*	Kaur et al., 2006 [33]
Filaments	Organogenesis; Somatic Embryogenesis	*R. hybrida* ‘Royalty’	Noriega and Söndahl, 1991 [74]
	Somatic Embryogenesis	*R. hybrida* ‘Royalty’	Firoozabady et al., 1994 [75]
		*R. hybrida* ‘Tournament of Roses’	Burrell et al., 2006 [31]
Ovules	Organogenesis; Somatic Embryogenesis	*R. hybrida* ‘Meirutral’	Arene et al., 1993 [72]
Petals	Organogenesis; Somatic Embryogenesis	*R. hybrida* ‘Meirutral’	Arene et al., 1993 [72]
	Somatic Embryogenesis	*R. hybrida* ‘Anny’	Borissova et al., 2000 [24]
Sepals	Organogenesis; Somatic Embryogenesis	*R. hybrida* ‘Meirutral’	Arene et al., 1993 [72]
Internodal Stem Segments	Organogenesis	*R. chinensis minima* ‘Baby Katie’ *R. chinensis minima* ‘Red Sunblaze’	Hsia and Korban, 1996 [61]
	Organogenesis; Somatic Embryogenesis	*R. hybrida* ‘Meirutral’	Arene et al., 1993 [72]
		*R. hybrida* ‘Charming’	Kim et al., 2004 [32]
	Somatic Embryogenesis	*R. hybrida* ‘Landora’	Rout et al., 1991 [70]
Leaflet Sections	Organogenesis	*R. hybrida* ‘Sonia’	Derks et al., 1995 [76]
		*R. hybrida* ‘Lavande’	Katsumoto et al., 2007 [25]
		*R. chinensis minima* ‘Baby Katie’ *R. chinensis minima* ‘Red Sunblaze’	Hsia and Korban, 1996 [61]
	Organogenesis; Somatic Embryogenesis	*R. hybrida* ‘Apollo’	Pourhosseini et al., 2013 [35]
		*R. hybrida* ‘Eiffel Tower’	Mahmoud et al., 2018 [21]
		*R. hybrida* ‘Carefree Beauty’	Hsia and Korban, 1996 [61]
		*R. hybrida* ‘Tineke’	Kim et al., 2004 [32]
	Somatic Embryogenesis	*R. hybrida* ‘Tournament of Roses’	Burrell et al., 2006 [31]
		*R. hybrida* ‘Livin’ Easy’	Estabrooks et al., 2007 [59]
		*R. chinensis* ‘Old Blush’	Cai et al., 2022 [20]
Petioles	Somatic Embryogenesis	*R. hybrida* ‘Glad Tidings’	Marchant et al., 1998 [77]
Leaflets	Organogenesis	*R. chinensis* ‘Old Blush’ *R. hybrida* ‘Delstrimen’ *R. hybrida* ‘Félicité et Perpétue’ *R. hybrida* ‘Natal Briar’ *R. hybrida* ‘White Pet’ *R. wichurana*	Hamama et al., 2019 [30]
	Organogenesis; Somatic Embryogenesis	*R. hybrida* ‘Heckenzauber’ *R. hybrida* ‘Pariser Charme’ *R. indica*	Dohm et al., 2001 [22]
		*R. hybrida* ‘Italian Ice’ *R. hybrida* ‘Ringo All-Star’ *R. hybrida* ‘Carefree Beauty’	Harmon et al., 2022 [78]
	Organogenesis; Protocorm-Like Bodies	*R. multiflora* var. *cathayensis* *R. multiflora* f. *carnea*	Tian et al., 2008 [49]
	Protocorm-Like Bodies	*R. canina*	Kou et al., 2016 [79]
	Somatic Embryogenesis	*R. chinensis* ‘Old Blush’	Cai et al., 2022 [20]
		*R. hybrida* ‘Domingo’ *R. hybrida* ‘Vickey Brown’	de Wit et al., 1990 [37]
		*R. hybrida* ‘Heckenzauber’	Dohm et al., 2001 [22]
		*R. hybrida* ‘Landora’	Das et al., 2010 [80]
		*R. hybrida* ‘Samantha’	Bao et al., 2012 [19]
Leaves	Organogenesis	*R. damascena* ‘Jwala’	Pati et al., 2004 [38]
		*R. hybrida* ‘Amanda’ *R. hybrida* ‘Black Baccara’ *R. hybrida* ‘Maroussia’ *R. hybrida* ‘Apollo’	Pourhosseini et al., 2013 [35]
		*R.* sp.	Nguyen et al., 2017 [23]
	Organogenesis; Somatic Embryogenesis	*R. hybrida* ‘Carefree Beauty’	Arene et al., 1993 [72]
		*R. hybrida* ‘Meirutral’	Li et al., 2002 [36]
		*R. hybrida* ‘Charming’ *R. hybrida* ‘4th of July’ *R. hybrida* ‘Tournament of Roses’	Kim et al., 2004 [32]
	Organogenesis; Somatic Embryogenesis; Secondary Somatic Embryogenesis	*R. chinensis* ‘Jacq.’	Chen et al., 2014 [58]
	Somatic Embryogenesis	*R. hybrida* ‘Landora’	Rout et al., 1991 [70]
		*R. hybrida* ‘Soraya’	Kintzios et al., 1999 [26]
		*R. hybrida* ‘Saltze Gold’	Borissova et al., 2000 [24]
		*R. hybrida* ‘Heckenzauber’ *R. hybrida* ‘Pariser Charme’	Dohm et al., 2001 [22]
		*R. hybrida* ‘Trimontsium’	Borissova et al., 2005 [81]
		*R. chinensis* ‘Jacq.’	Chen et al., 2010 [65]
		*R. chinensis* ‘Old Blush’	Vergne et al., 2010 [69]
		*R. hybrida* ‘Linda’	Zakizadeh et al., 2013 [82]
		*R. hybrida* ‘E.H.L.Krause RI	Randoux et al., 2014 [66]
		*R. hybrida* ‘Samantha’	Liu et al., 2021 [15] Wang et al., 2023 [17]
		*R. hybrida* ‘Carola’	Duan et al., 2024 [67]
	Somatic Embryogenesis; Secondary Somatic Embryogenesis	*R. hybrida* ‘Carefree Beauty’	Li et al., 2002 [36]
Roots	Organogenesis; Somatic Embryogenesis	*R. hybrida* ‘Meirutral’	Arene et al., 1993 [72]
		*R. hybrida* ‘Charming’	Kim et al., 2004 [32]
	Somatic Embryogenesis	*R. persica* × *xanthina*	Matthews et al., 1991 [63]
		*R. hybrida* ‘Fresham’	Yokoya et al., 1996 [71]
Not Specified	Organogenesis; Somatic Embryogenesis	*R. hybrida* ‘Pariser Charme’ *R. persica* × *R. xanthina* *R. hybrida* ‘Elina’	Schum et al., 2001 [64]
	Somatic Embryogenesis	*R. floribunda* ‘Trumpeter’ *R. hybrida* ‘Dr. Huey’ *R. multiflora* ‘Tineké’	Castillón and Kamo, 2002 [34]

### 6.4. Media Composition

Tissue culture media is formulated based on the plant species, developmental stage, and desired plant response. Various basal media, growth regulators, and additives have been used in rose tissue culture and regeneration (Table 2). The basal medium provides macronutrients, micronutrients, and vitamins. Meanwhile, hormone additives, typically mixed after autoclaving of the core media components, include auxins and cytokinins that regulate cell division, differentiation, and morphogenesis. A high auxin-to-cytokinin ratio favors root formation, while a low ratio promotes shoot development. Murashige and Skoog (MS) basal salts are the predominant basal salt formulation and support various regeneration types, including somatic embryo formation, in numerous *R. hybrida* cultivars such as ‘Tournament of Roses’, ‘Carola’, and ‘Black Baccara’ [31,34,72]. MS basal salts are also commonly used for closely related species like *R. rugosa* and *R. chinensis* [57,73]. While less frequent, Gamborg’s B5 and Schenk and Hildebrandt (SH) basal salts are also utilized and can be more successful in inducing regeneration for certain genotypes [58,74]. The media often has additional Iron(III) ethylenediamine-N,N′-bis(2-hydroxyphenylacetic) acid (Fe-EDDHA) applied to ensure optimal iron uptake [30]. Vitamin supplementation represents an important yet variable component in culture media, with different formulations often influencing regeneration outcomes. Among them, MS vitamins are the most widely used, typically supplied together with MS basal salts in most reported protocols [23,75]. Other vitamin solutions, including B5 and SH vitamins, are reported to be compatible with regeneration in roses [31,71]. Fewer studies mentioned the use of specialized vitamin solutions, such as Morel and Martin vitamins and Staba vitamins [27,76]. Many vitamin formulations contain additional myo-inositol, nicotinic acid, and various amino acids to improve regeneration response [66], while other medium recipes may add these separately.

In addition, carbohydrate sources provide the energy required for *in vitro* growth. Sucrose is the predominant carbohydrate reported to consistently support various regeneration types, including both organogenic shoots and somatic embryo formation, across numerous rose cultivars [20,38,59]. However, glucose, fructose, and maltose have been used to replace or supplement sucrose in multiple reports, sometimes yielding superior results for genotypes such as *R. hybrida* ‘Carefree Beauty’ and ‘Carola’ [67,75]. Less common sugar alcohols like mannitol and sorbitol, which may help the osmotic balance, also appeared in successful protocols [20]. These sugars are commonly used to improve the germination percentage of somatic embryos [30].

**Table 2 plants-14-03797-t002:** Reported media compositions and conditions for successful rose regeneration pathways.

Media Composition	Associated PGRs	Environmental Conditions	Associated Publications
MS Modified vitamins Fructose; Glucose; Maltose; Sucrose	BAP; Fe-EDDHA; IBA; TDZ	21 ± 2 °C Dark conditions 16-h photoperiod	Hamama et al., 2019 [30]
MS Modified vitamins Sucrose	GA_3_; NAA; Zeatin	24 ± 1 °C Dark conditions	Noriega and Söndahl, 1991 [74]
	BAP; IAA	25 ± 1 °C Dark conditions	Firoozabady et al., 1994 [75]
	2,4-D	25 °C Dark conditions	Castillón and Kamo, 2002 [34]
	Dicamba; IBA; Kinetin	25 ± 1 °C Dark conditions 16-h photoperiod	Kim et al., 2004 [32]
	2,4-D; BAP; GA_3_	25 ± 2 °C 16-h photoperiod	Das et al., 2010 [80]
MS Morel and Martin Vitamins Sucrose	2,4-D; BAP	23 °C 16-h photoperiod	Arene et al., 1993 [72]
MS Morel and Wetmore Vitamins Sucrose	AgNO_3_; BAP; GA_3_; NAA	25 ± 1 °C 16-h photoperiod	Mahmoud et al., 2018 [21]
MS MS Vitamins Fructose; Glucose; Maltose; Sorbitol; Sucrose	2,4-D; Adenine; BAP; Kinetin; NAA; Picloram; Zeatin	25 °C Continuous photoperiod	Kunitake et al., 1993 [57]
MS MS Vitamins Fructose; Glucose; Sucrose	2,4,5-T; 2,4-D; BAP; TDZ	23 °C Dark conditions Photoperiod (not described)	Harmon et al., 2022 [78]
MS MS Vitamins Glucose	2,4-D; BAP; TDZ	24 ± 2 °C Dark conditions 14 h photoperiod	Bao et al., 2012 [19]
	AgNO_3_; BAP; IBA; TDZ	23 ± 2 °C Dark conditions 16-h photoperiod	Nguyen et al., 2017 [23]
	2,4-D; BAP; GA_3_; Kinetin; NAA	22 ± 2 °C Dark conditions 16-h photoperiod	Liu et al., 2021 [15]
	BAP; IBA; NAA; Zeatin	25 ± 1 °C Dark conditions 16-h photoperiod	Duan et al., 2024 [67]
MS MS Vitamins Mannitol; Sucrose	GA_3_; IBA; NAA; Zeatin	25 °C Dark conditions light conditions	Vergne et al., 2010 [69]
MS MS Vitamins Sucrose	Kinetin; NAA	20 °C Dark conditions 16-h photoperiod	de Wit et al., 1990 [37]
	2,4-D; Adenine sulfate; BAP; GA_3_; NAA	25 ± 2 °C; 8 ± 1 °C Dark conditions 16-h photoperiod	Rout et al., 1991 [70]
	GA_3_; TDZ	22 ± 1 °C Dark conditions 16-h photoperiod	Hsia and Korban, 1996 [61]
	BAP; IAA; Kinetin; pCPA	25 °C 16-h photoperiod	Kintzios et al., 1999 [26]
	2-iP; Dicamba; Kinetin	25 °C Dark conditions 16-h photoperiod	Borissova et al., 2000 [24]
	GA_3_; NAA; Zeatin	25 °C Ambient photoperiod (not described) 90 rpm 23 ± 2 °C Dark conditions 16-h photoperiod	Dohm et al., 2001 [22]
	BAP; NAA	25 °C Photoperiod (not described)	Asano and Tanimoto, 2002 [73]
	2,4-D	25 °C 16-h photoperiod	Kim et al., 2004 [32]
	2,4-D; Zeatin	25 °C Dark conditions 16 h photoperiod	Kim et al., 2004 [32]
	AgNO_3_; BAP; NAA; TDZ	25 ± 2 °C Dark conditions 14 h photoperiod	Pati et al., 2004 [38]
	2-iP; 2,4-D; ABA; BAP; Dicamba; GA_3_; Kinetin	25 °C Dark conditions 16-h photoperiod	Borissova et al., 2005 [81]
	2,4-D; BAP	25 ± 2 °C Dark conditions 16-h photoperiod	Kaur et al., 2006 [33]
	2,4,5-T; ABA	22 °C Dark conditions	Estabrooks et al., 2007 [59]
	BAP	25 °C 16-h photoperiod	Katsumoto et al., 2007 [25]
	2,4-D; BAP; GA_3_; IBA; TDZ	25 ± 2 °C Dark conditions 16-h photoperiod	Tian et al., 2008 [49]
	2,4-D; BAP; GA_3_; IBA; Kinetin; NAA; NPA; TDZ	25 ± 2 °C Dark conditions 16-h photoperiod	Kou et al., 2016 [79]
	2,4-D; GA_3_; TDZ	23 ± 1 °C Dark conditions 16-h photoperiod	Li et al., 2002 [36]
	2,4-D; GA_3_;TDZ 2,4-D; ABA; BAP; GA_3_; NAA; TDZ; Zeatin	22 ± 2 °C Dark conditions	Pourhosseini et al., 2013 [35]
		23 ± 2 °C Dark conditions 16-h photoperiod	Zakizadeh et al., 2013 [68]
MS Staba Vitamins Glucose; Sucrose	2,4-D; BAP	Not described	Randoux et al., 2014 [66]
	2,4-D; Zeatin	24 ± 3 °C Dark conditions	Burrell et al., 2006 [31]
MS; B5 MS Vitamins; B5 Vitamins Glucose; Sucrose	2,4-D; BAP; IBA; NAA	22 °C Dark conditions Photoperiod (not described)	Derks et al., 1995 [76]
MS; SH Modified vitamins Sucrose	2,4-D; BAP; GA_3_; Kinetin	25 ± 2 °C 16-h photoperiod	Cai et al., 2022 [20]
MS; SH MS Vitamins; SH Vitamins Sucrose	2,4-D	25 °C Dark conditions 16-h photoperiod	Schum et al., 2001 [64]
Not described	2,4-D	25 °C Dark conditions 130 rpm	Wang et al., 2023 [17]
		25 °C Dark conditions Photoperiod (not described)	Matthews et al., 1991 [63]
SH Modified vitamins Sucrose	2,4-D; BAP; NAA	25 °C Dark conditions 16-h photoperiod	Kim et al., 2004 [32]
	2,4-D; TDZ	24 °C; 4 °C Dark conditions Photoperiod (not described)	Yokoya et al., 1996 [71]
SH SH Vitamins Maltose; Sucrose	2,4-D; ABA; GA_3_; TDZ	25 ± 2 °C Dark conditions 16-h photoperiod	Chen et al., 2010 [65]
SH SH Vitamins Sucrose	2,4-D; ABA; GA_3_; TDZ	25 ± 2 °C Dark conditions 16-h Photoperiod (not described); red light; white light	Chen et al., 2014 [58]

### 6.5. Plant Growth Regulators (PGRs)

PGRs are organic/inorganic compounds that significantly influence and modify plant growth, development, and physiological processes. Auxins are fundamental for inducing callus formation, promoting root development, and initiating somatic embryo formation [77,78]. The most reported auxin used to induce regenerative callus across multiple genotypes in rose is 2,4-dichlorophenoxyacetic acid (2,4-D). The concentration of 2,4-D added to the media between 2.26 µM and 24.88 µM. 2,4-D was effective in inducing embryogenic callus in *R. hybrida* ‘Carefree Beauty’, ‘Dr. Huey’, ‘4th of July’, ‘Tournament of Roses’, *R. floribunda* ‘Trumpeter’, and *R. multiflora* ‘Tineké’ [34,36,79]. 2,4-D was also successful at inducing callus from which organogenic shoots later developed in *R. hybrida* ‘Meirutral’, *R. multiflora* var. *cathayensis*, *R. multiflora* f. *carnea*, and *R. canina* [27,49,80]. Other synthetic auxins like 2,4,5-trichlorophenoxyacetic acid (2,4,5-T) and 3,6-dichloro-2-methoxybenzoic acid (Dicamba) have also been reported in research papers to aid in the development of regenerative plant tissues [32,81]. For example, *R. hybrida* ‘Livin’ Easy’ produced somatic embryos after callus was cultured on a medium containing 25 µM 2,4,5-T [59]. *R. hybrida* ‘Anny’ and ‘Saltze Gold’ produced somatic embryos when the plant material was treated with a combination of 2,4-D and Dicamba [82]. In contrast, while 4-amino-3,5,6-trichloropyridine-2-carboxylic acid (Picloram) is widely employed for regeneration in many monocots, its use in rose tissue culture is rarely documented, with only two reported successful instances. A combination of 2,4-D and Picloram was successful at initiating embryogenic callus in *R. rugosa* Thunb., while *R. hybrida* ‘Pariser Charme’ produced somatic embryos and organogenic shoots when treated with Picloram [57,64]. While 4-Chlorophenoxyacetic acid (pCPA) has also demonstrated the capacity to induce somatic embryos in roses, recorded occurrences remain infrequent. *R. hybrida* ‘Soraya’ produced somatic embryos on callus induced on media containing 53.5 μM pCPA, but *R. hybrida* ‘Ronto’ did not produce any regenerative features with the same treatment [26]. Auxins like naphthaleneacetic acid (NAA), indole-3-acetic acid (IAA), and indole-3-butyric acid (IBA) were documented to regulate shoot production or induce roots from regenerated shoots. *R. hybrida* ‘Heckenzauber’ developed embryogenic callus on media supplemented with 1.34 µM of NAA as well as multiple cytokinins [22]. IBA is commonly used to induce roots in *in vitro* rose plants before starting the acclimation process [23,67].

Cytokinins are a class of hormones that promote cell division and shoot regeneration/proliferation. Cytokinins often have synergistic or antagonistic relationships with other classes of hormones, and so are often added to media in conjunction with other signaling compounds [19]. The most common cytokinin used in tissue culture is 6-Benzylaminopurine (BAP), which induces organogenic shoots and somatic embryo germination across multiple rose genotypes (typically delivered between 1.11 and 8.88 μM). BAP was added at 4.44 and 5.0 μM, respectively, to germinate somatic embryos in *R. rugosa* ‘Bao White’ and *R. bourboniana* [36]. Kinetin and zeatin are other cytokinins that support various regeneration pathways. *R. hybrida* ‘Domingo’ and ‘Vickey Brown’ produced somatic embryos when treated with 0.46 μM kinetin [37]. Thidiazuron (TDZ) is a synthetic cytokinin noted for its strong regenerative capabilities, especially in inducing somatic embryos from callus. *R. hybrida* ‘Linda’ produced somatic embryos when treated with 45.4 μM of TDZ [68]. Similarly, *R. hybrida* ‘Carefree Beauty’ produced organogenic shoots and somatic embryos on callus treated with 2.3 μM of TDZ [36]. A low concentration of TDZ (2.3–9.6 μM) was also capable of producing organogenic shoots in *R. wichurana*, *R. hybrida* ‘White Pet’, *R. chinensis* ‘Old Blush’, *R. hybrida* ‘Delstrimen’, and *R. hybrida* ‘Félicité et Perpétue’ [30]. Other growth regulators like abscisic acid and gibberellic acid also affect embryo development. For example, the addition of 0.3 μM gibberellic acid promoted shoot elongation and embryo maturation in *R. hybrida* ‘Landora’ [70]. Abscisic acid improved somatic embryo maturation in *R. hybrida* ‘Leonie’, ‘Eiffel Tower’, and ‘Carefree Beauty’ [21,36,68].

## 7. Organic and Inorganic Additives

Both organic and inorganic compounds play important roles in regulating somatic embryogenesis in roses. Among organic additives, casein hydrolysate, coconut milk, and malt extract are commonly used because they provide a wide range of nutrients and growth-promoting factors that help support tissue regeneration [83]. Casein hydrolysate has shown a strong positive effect: when added to Vander Salm medium, it stimulated shoot regeneration in *R. canina*, increasing multi-shoot induction by 173% compared with the control treatment [84]. Somatic embryo formation was promoted by malt extract in *R. rugosa* [57]. In addition to nutrient-rich supplements, certain compounds were added to culture media to counteract the negative effects of phenolic exudation. Activated charcoal, cysteine, and ascorbic acid were often used for this purpose. For example, embryogenic callus derived from F1 embryos of *R. hybrida* ‘Bridal Pink’ successfully produced organogenic shoots on medium supplemented with both ascorbic acid (5.7 μM) and cysteine (8.3 μM) [62]. Likewise, activated charcoal has been reported to promote embryo development; the addition of 0.25 M charcoal was particularly effective for inducing somatic embryos in *R. hybrida* ‘Anny’ and ‘Saltze Gold’ [24].

## 8. Case Studies

### 8.1. ‘Samantha’

*R. hybrida* ‘Samantha’ is a model system in rose plant biotechnology. Multiple studies have shown successful regeneration through somatic embryo formation from callus. One study utilized expanded leaves, while two others focused on leaflets with petioles as explant sources. ‘Samantha’ produced embryogenic callus when the explants were placed on a medium composed of MS, glucose, and a combination of PGRs, including 2,4-D [15,17,72]. This consistent regeneration provides a reliable foundation for downstream genetic transformation work. ‘Samantha’ has been successfully transformed via *Agrobacterium*-mediated methods to express *GFP* and a hygromycin B resistance marker [72]. Its receptiveness to advanced techniques is evidenced by its successful application in a CRISPR/Cas9 system for targeted genome editing [17].

### 8.2. ‘Carefree Beauty’

*R. hybrida* ‘Carefree Beauty’ is another model in rose biotechnology. The cultivar’s regenerative capacity is well-established, with protocols developed for multiple explant sources. Studies showed that whole leaf explants can produce somatic embryo formation from leaf callus, and shoot organogenesis from petiole callus [36]. This dual regenerative capacity was also observed in an earlier study using leaf sections, where somatic embryos originated from callus [61]. Later work further confirmed that leaflets can produce callus capable of both organogenesis and somatic embryo formation [78]. ‘Carefree Beauty’ has been successfully transformed using *Agrobacterium tumefaciens* with the *uidA* gene [85]. The antimicrobial protein gene *Ace-AMP1* was also introduced into the cultivar to confer resistance to powdery mildew [86]. 

### 8.3. ‘Rosaceae Crops’

Rosaceae is the third most economically important plant family in temperate regions. The high economic value has driven significant advancements in tissue culture regeneration and biotechnology in rosaceous crops. Among rosaceous species, apples and strawberries are the most extensively investigated, particularly with respect to efficient organogenesis and somatic embryogenesis. In apple, leaf regeneration can be achieved via two major approaches. One method uses an initial dark treatment with 2,4-D and NAA to induce embryogenic cells, followed by light culture with BA to promote efficient somatic embryogenesis. Alternatively, shoot organogenesis can be induced directly by culturing leaf explants on MS medium supplemented with BA alone or combined with NAA in the dark. Regeneration capacity varies by genotype [87,88], with ‘Gala’ and ‘Greensleeves’ among the most responsive cultivars [87,89,90]. Wounding the leaf explant prior to culture improved both the regeneration rate and the number of regenerated shoots in the apple variety ‘Gala’ [91]. Strawberry, widely used as a model for tissue culture and biotechnological studies. Successful regeneration has been achieved from a variety of explants, including leaf, stipule, petiole, roots, and stems. For transformation studies, however, leaf disks remain the predominant explant type [92]. Effective concentrations of growth regulators typically include 4.44–13.32 μM BA or TDZ, supplemented with 0–1.07 μM NAA or 0–1.45 μM GA_3_ [93,94]. As in apple, substantial cultivar-to-cultivar variation exists; Festival’ and ‘Chandler’ show particularly high regeneration capacities [93,95]. Explant source also influences regeneration outcomes, as high-frequency adventitious shoot regeneration has been reported from petiole explants [96].

## 9. Future Prospects

Plant regeneration is a complex process in which internal and external factors act in concert to achieve final plantlet regeneration. Internally, broad genotype screening is required to assess the regeneration potential of different varieties. Immature tissue and seed-derived explants are generally more responsive than mature and vegetative tissues, and deeper insight into the molecular mechanisms underlying rose regeneration will be essential for developing genotype-independent protocols. Externally, regeneration efficiency depends heavily on the composition of the induction medium, the specific combinations of growth regulators, and culture environmental conditions, all of which require continual optimization. Moreover, incorporating a cell suspension culture step during callus selection has been shown to enhance embryogenic cell enrichment and improve regeneration outcomes [15,16].

Recent advances in genotype-independent transformation and *in planta* transformation offer promising opportunities to overcome the long-standing recalcitrancy of rose tissue culture regeneration, thereby enabling new strategies for rose biotechnology and breeding. Morphogenetic factors (MFs) are specialized plant genes that play pivotal roles in plant growth and meristem development. Overexpression of MFs such as *BBM*, *WUS*, and the pair of GROWTH REGULATING FACTOR4 AND GRF-INTERACTING FACTORS (*GRF4*-*GIF1*) complex has markedly enhanced somatic embryogenesis in recalcitrant crops and cultivars, enabling genotype-independent regeneration and transformation in species such as cotton, maize, and rice [97]. To fully harness these MFs for rose transformation and regeneration, further refinement of regeneration protocols-particularly reducing reliance on high levels of exogenous growth regulators will be necessary.

*In planta* transformation methods, including floral dip and meristem bombardment transformation, provide alternative strategies for generating transgenic roses. *Arabidopsis* floral dip is a tissue culture-independent transformation approach in which inflorescences are immersed in an *Agrobacterium tumefaciens* suspension carrying the construct of interest [98]. Transgenic seeds are subsequently identified using chemical or visual selection markers. This strategy may apply to rose genotypes that are miniature in stature and capable of producing abundant seeds. *In planta* meristem bombardment offers another approach that requires minimal tissue culture. For example, in wheat, isolated embryos are bombarded with gold particles coated with a GFP construct, and transgenic plants are recovered by simply germinating the bombarded embryos, bypassing callus formation and regeneration steps [99]. This method may be feasible for rose genotypes that possess a mature *in vitro* propagation system but lacks an efficient tissue culture-based regeneration pathway.

These integrative approaches, together with advances in rose tissue culture, molecular and stress physiology, will provide powerful strategies for developing novel rose cultivars with superior traits, even in rose genotypes that are highly recalcitrant to regeneration.

## Figures and Tables

**Figure 1 plants-14-03797-f001:**
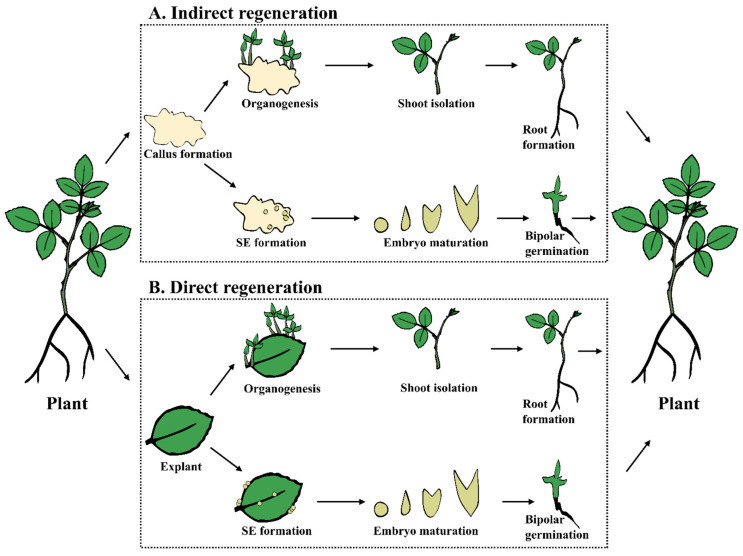
Primary pathways for rose tissue culture regeneration. (**A**) Indirect regeneration is characterized by a callus stage before organogenesis or somatic embryogenesis and later plantlet formation. (**B**) Direct regeneration happens directly on explants (leaf, as an example here) without a callus stage.

**Figure 2 plants-14-03797-f002:**
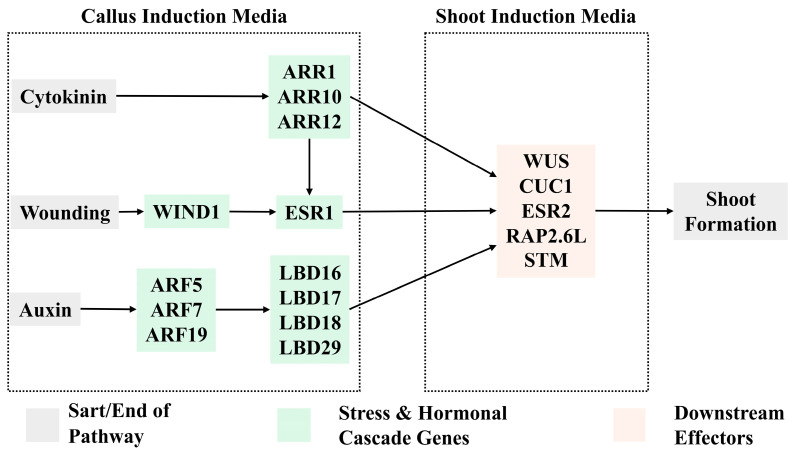
Molecular pathway for shoot organogenesis. Wounding triggers the *WIND1* pathway. Transferring plant material to a callus induction medium (CIM) containing cytokinins upregulates *ARR1* and *ARR2*, while auxin application upregulates *ARFs* and *LBDs*. Continual upregulation of *LBDs* and *ESR1* in conjunction with a transfer to a shoot induction media (SIM) high in cytokinin results in the upregulation of *WUS*, *CUC2*, *ESR2*, *STM*, and *RAP2.6L*.

**Figure 3 plants-14-03797-f003:**
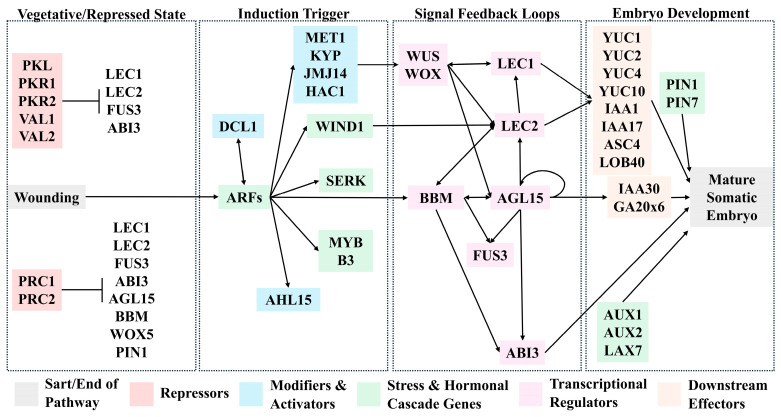
Molecular pathway for somatic embryo formation. Exogenous environmental signals trigger the activation of ARFs, which activate the expression of *WUS*/*WOX*, *WIND1*, *BBM*, and *SERK* genes. *WUS*/*WOX*, *WIND1*, and *BBM* all trigger activation of the LAFL genes, leading to embryo formation and maturation.

**Figure 4 plants-14-03797-f004:**
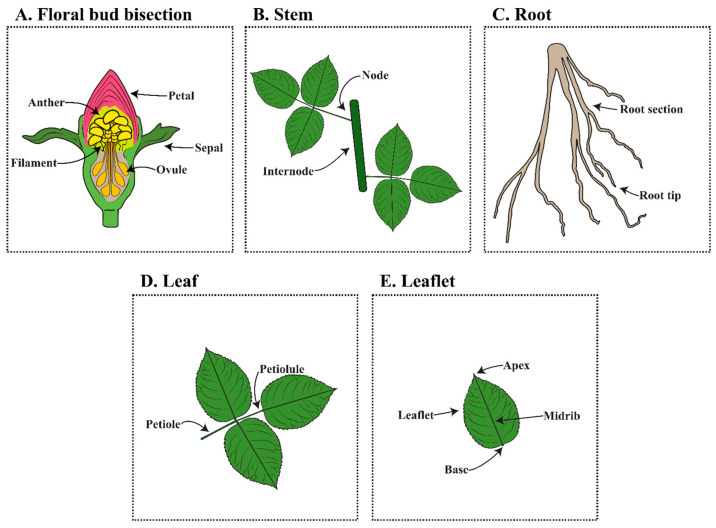
Explants used for rose tissue culture regeneration. (**A**) Floral bud including anther, petal, sepal, filament, and ovule. (**B**) Stem, including the stem node and internode. (**C**) Root, including the root section and root tip. (**D**) Trifoliar leaf, including the leaf with petiole and petiole. (**E**) Trifoliar leaflet.

## Data Availability

No new data were created or analyzed in this study. Data sharing is not applicable to this article. Original figures, tables, and images are available from the corresponding author upon reasonable request.

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
