# Peer review of "Overcoming Recalcitrance: A Review of Regeneration Methods and Challenges in Roses"

_plants, 2025, doi:10.3390/plants14243797_

Round 1
Reviewer 1 Report
Comments and Suggestions for Authors
Overall, this is a well-written review article. The topic is of interest to the research community, and the content is well-thought out. I would like to point out a few content- and stylistic- issues that I noted in this manuscript, and so I hope the authors will take note of these issues and make appropriate changes/modifications.
- As the title of this manuscript clearly state that this review is focused on the issue of "recalcitrance", I would have like to see additional discussion of this topic both in the introduction section as well as in the latter sections of the manuscript. It would be informative and useful to include how recalcitrance in rose compares to that reported in other plant species, and whether or not some of the approaches used in other plant species to overcome recalcitrance may or may not aid in overcoming this problem in rose. It would be useful to also include a section on any cellular or molecular factors that have been identified in other recalcitrant plant species that might also be informative and helpful to overcome recalcitrance in rose. You may choose to include this either early in the body of the manuscript and/or also in the last section of the manuscript. I will leave that decision up to you.
- Once the genus name, Rosa, is written in full for the first time for the different rose species in the body of the manuscript, the authors should simply use the abbreviation, R., for the remainder of the text. Likewise, once the full name of the genus is written in full in each of the tables, subsequently, use the abbreviation, R., for the remainder of that table.
- I have noted that some of the species are not italicized, so please correct this issue throughout the manuscript, both in the body of the text and in all references.
- In the body of the text, the reference of "Hsia and Korban" is dated for the year "1996"; however, in the text this cited reference is written as "1995". Please correct the date in the table.
- In the last section of this manuscript, it will be useful to include advances in genome-wide studies in rose that cover both biological and environmental factors that might shed some light on the topic of recalcitrance.
Author Response
Response to reviewer’s suggestions:
Reviewer 1:
- As the title of this manuscript clearly state that this review is focused on the issue of "recalcitrance", I would have like to see additional discussion of this topic both in the introduction section as well as in the latter sections of the manuscript. It would be informative and useful to include how recalcitrance in rose compares to that reported in other plant species, and whether or not some of the approaches used in other plant species to overcome recalcitrance may or may not aid in overcoming this problem in rose. It would be useful to also include a section on any cellular or molecular factors that have been identified in other recalcitrant plant species that might also be informative and helpful to overcome recalcitrance in rose. You may choose to include this either early in the body of the manuscript and/or also in the last section of the manuscript. I will leave that decision up to you.
Two comparative studies were added including molecular and biochemical factors related to rose regeneration. We also include two Rosaceae crop regeneration progress in comparison to rose in the ‘Case study’ section.
- Once the genus name, Rosa, is written in full for the first time for the different rose species in the body of the manuscript, the authors should simply use the abbreviation, R., for the remainder of the text. Likewise, once the full name of the genus is written in full in each of the tables, subsequently, use the abbreviation, R., for the remainder of that table.
Good suggestions. All corrected following suggestions.
- I have noted that some of the species are not italicized, so please correct this issue throughout the manuscript, both in the body of the text and in all references.
Thanks for your suggestions. All corrected following suggestions.
- In the body of the text, the reference of "Hsia and Korban" is dated for the year "1996"; however, in the text this cited reference is written as "1995". Please correct the date in the table.
Thanks for pointing out this. It is corrected and the right year is 1996
- In the last section of this manuscript, it will be useful to include advances in genome-wide studies in rose that cover both biological and environmental factors that might shed some light on the topic of recalcitrance.
We added Biochemistry of rose regeneration and highlighted comparative genomics importance in on rose regeneration study. Pointing out the potential application of morphological factors and in planta transformation methodology in future rose biotechnology breeding
Reviewer 2 Report
Comments and Suggestions for Authors
Dear,
The review paper entitled “Overcoming Recalcitrance: A Review of Regeneration Methods and Challenges in Rose” is an interesting topic for exposing the potent methods and challenges in Rose. The article is written in a good format but needs necessary major revision for further refinement.
- Paragraphs are written in continuous form. Please rearrange the paragraphs in entire sections of the manuscript.
- Some lines and pragraphs are continuing and ending in an irregular style on a new line. E.g., line 38. Check these mistakes in the entire manuscript.
- Please add more comparative studies on gene regulation and molecular, physiological, and histological attributes.
- Add more case studies related to different crops of the Rosaceae family. It will provide the comparative validation for the readers.
- Please add the detailed schematic diagram illustrating genes and regulatory pathways involved in the regeneration activities in leaves, petioles, and internodes.
The English could be improved to more clearly express the research.
Author Response
Response to reviewer 2:
The review paper entitled “Overcoming Recalcitrance: A Review of Regeneration Methods and Challenges in Rose” is an interesting topic for exposing the potent methods and challenges in Rose. The article is written in a good format but needs necessary major revision for further refinement.
- Paragraphs are written in continuous form. Please rearrange the paragraphs in entire sections of the manuscript.
Accepted suggestions and rearranged the paragraphs.
- Some lines and paragraphs are continuing and ending in an irregular style on a new line. E.g., line 38. Check these mistakes in the entire manuscript.
Accepted suggestions and rearranged the paragraphs.
- Please add more comparative studies on gene regulation and molecular, physiological, and histological attributes.
Gene regulation on molecular (131) and biochemical levels (220) was added. Comparative genomics application in rose research was added(193-202).
- Add more case studies related to different crops of the Rosaceae family. It will provide the comparative validation for the readers.
A comparative study of two Rosaceae crops (apple and strawberry) regeneration progress was added to the ‘Case study’ section.
- Please add the detailed schematic diagram illustrating genes and regulatory pathways involved in the regeneration activities in leaves, petioles, and internodes.
We focus on illustrating internal and external factors affecting rose explant regeneration. A schematic diagram was added illustrating genes and regulatory pathways involved plant regeneration (include leaf, petiole and internodes); A paragraph of biochemical indicators related to rose regeneration was added.
Reviewer 3 Report
Comments and Suggestions for Authors
Title
Overcoming Recalcitrance: A Review of Regeneration Methods and Challenges in Rose
The manuscript is in a different format (Cells Journal).
Abstract
Lines 10-12: Why are these techniques required?
Lines 12-14: As with most plant species. I would suggest improving the justification; that is, all in vitro cultivated plants present these difficulties, however, what specifically happens in rose?
Lines 19-21: The novelty of this work needs to be emphasized more.
In general, the abstract is well-structured; however, the novelty of this review compared to existing ones needs to be mentioned.
Keywords
Introduction
References should be listed in order of appearance and in brackets.
Line 38: Period.
Lines 46-48: Mention why? Or at least investigate according to the literature. Lines 52-53 above mention the long juvenile period of roses and the environmental requirements for a rose to flower. It seems to me that tissue culture does not solve these challenges.
How does this address the above?
The information needs to be organized. Plant tissue culture is mentioned as a solution to all the problems that exist in rose cultivation; however, there is a further problem: the rooting and development of a rose that comes from PTC (Public Tissue Culture). There is also the recalcitrant nature of some genotypes.
- Challenges and Limitations in Rose Tissue Culture Regeneration
Lines 81-84 (please pay attention to the formatting). Is the idea focused on rose cultivation? Remember that we have already covered the general limitations that occur in PTC, and now we are focusing specifically on what happens in roses.
The title of this section mentions the limitations in rose PTC; however, much of the information presented applies to other crops. To do this, a general section on the limitations of rose tissue culture (other crops or in general) should be included, followed by a specific section (the rose case).
- Methods of Rose Regeneration in Tissue Culture
Lines 116-118: This information is not in the rose section.
This section should be organized: first, explain what morphogenesis is; then, discuss organogenesis and indirect embryogenesis; and finally, discuss organogenesis and indirect embryogenesis (as shown in the figure). Explain what each entails and how it has been addressed in the rose section.
- Advances and Trends in Rose Tissue Culture (1990–2025): Advances in Regeneration Techniques
4.1. Environmental Conditions – Temperature
Lines 159-162: This idea needs bibliographic support (cite).
Lines 178-180: Are you referring to indirect organogenesis? Callus-shoots.
An interpretation of all the described information by the authors is needed; a review is not simply listing study after study, but interpreting the information and providing a recommendation based on what has been presented.
4.2. Environmental Conditions – Liquid Cultures
What was the criterion for ordering the information? The date of preparation was not used.
4.3. Explant Source and Selection
Lines 25-27.
The table should be ordered according to a criterion, such as the date of the study in studies that share sections, or perhaps the alphabetical order of the cultivars.
The expected advantages of each type of explant should be mentioned.
4.4. Media Composition
Lines 251-283 should be ordered according to the component analyzed, for example, salts, carbon source, organic compounds, etc.
Table 2 does not have the same format as Table 1.
4.5. Plant Growth Regulators
Table 2 can be mentioned.
- Organic and inorganic additives
- Case Study
Describe why these cultivars are a model system, what their characteristics are, and why other cultivars are not.
- Future Prospects and Applications
Lines 402-406 How was this conclusion reached? This topic was not addressed throughout the text, except superficially in the introduction.
The interpretation of the direction of PTC research in roses should be based on all the information compiled by the authors. Which technique has yielded the best results, and why? What is missing? How much progress has been made in addressing what is missing? Where should research be directed? The ideas expressed in this section can be suggested even without reading the text.
Author Response
Reviewer 3:
Overcoming Recalcitrance: A Review of Regeneration Methods and Challenges in Rose
The manuscript is in a different format (Cells Journal).
Corrected to Plants
Abstract
Lines 10-12: Why are these techniques required?
We re-organized this part and focused on rose biotechnology breeding needs for a high efficient regeneration system.
Lines 12-14: As with most plant species. I would suggest improving the justification; that is, all in vitro cultivated plants present these difficulties, however, what specifically happens in rose?
We re-organized this part and focused on low regeneration frequency and strong genotype dependency are the main limitations in rose biotechnology breeding.
Lines 19-21: The novelty of this work needs to be emphasized more.
The underlying causes of recalcitrance are discussed in relation to tissue physiology, biochemical and molecular regulation of morphogenesis. Finally, strategies for overcoming regeneration barriers—such as the use of morphogenic regulators and in planta transformation—are highlighted as emerging avenues toward genotype-independent rose genetic transformation and genome editing.
In general, the abstract is well-structured; however, the novelty of this review compared to existing ones needs to be mentioned.
We re-organized the abstract and put focus on biotech breeding needs for an efficient regeneration system.
Keywords
Introduction
References should be listed in order of appearance and in brackets. Corrected
Line 38: Period. Corrected
Lines 46-48: Mention why? Or at least investigate according to the literature. Lines 52-53 above mention the long juvenile period of roses and the environmental requirements for a rose to flower. It seems to me that tissue culture does not solve these challenges. How does this address the above?
We re-organized this part and focused on low regeneration frequency and strong genotype dependency are the main limitations in rose biotechnology breeding.
The advancement of novel rose cultivars with traditional methods are hindered by factors such as reproductive barriers [7,8], extended generation intervals [9], and polyploidy [10,11,12]. In contrast, genetic transformation and genome editing enable the introduction of desired traits into elite varieties in a short period of time.
The information needs to be organized. Plant tissue culture is mentioned as a solution to all the problems that exist in rose cultivation; however, there is a further problem: the rooting and development of a rose that comes from PTC (Public Tissue Culture). There is also the recalcitrant nature of some genotypes.
We re-organized this part and focused on low regeneration frequency and strong genotype dependency are the main limitations in rose biotechnology breeding.
- Challenges and Limitations in Rose Tissue Culture Regeneration
Lines 81-84 (please pay attention to the formatting). Is the idea focused on rose cultivation? Remember that we have already covered the general limitations that occur in PTC, and now we are focusing specifically on what happens in roses.
The title of this section mentions the limitations in rose PTC; however, much of the information presented applies to other crops. To do this, a general section on the limitations of rose tissue culture (other crops or in general) should be included, followed by a specific section (the rose case).
Thank you for suggestions, we updated this part and made it focused on challenges and limitations rose regeneration faces.
- Methods of Rose Regeneration in Tissue Culture
Lines 116-118: This information is not in the rose section.
This section should be organized: first, explain what morphogenesis is; then, discuss organogenesis and indirect embryogenesis; and finally, discuss organogenesis and indirect embryogenesis (as shown in the figure). Explain what each entails and how it has been addressed in the rose section.
Thank you for your suggestions. We reorganized this part and made it clear for rose regeneration path way models.
- Advances and Trends in Rose Tissue Culture (1990–2025): Advances in Regeneration Techniques
4.1. Environmental Conditions – Temperature
Lines 159-162: This idea needs bibliographic support (cite).
Corrected. We list all various environmental conditions for regeneration in table 2 and add table 2 to this part.
Lines 178-180: Are you referring to indirect organogenesis? Callus-shoots.
Corrected to callus-shoots
An interpretation of all the described information by the authors is needed; a review is not simply listing study after study, but interpreting the information and providing a recommendation based on what has been presented.
Discussion and recommendation were added to correspondent parts.
4.2. Environmental Conditions – Liquid Cultures
What was the criterion for ordering the information? The date of preparation was not used.
Liquid culture is important for embryogenic cell proliferation and selection reported by several papers, publication date was added.
4.3. Explant Source and Selection
Lines 25-27.
The table should be ordered according to a criterion, such as the date of the study in studies that share sections, or perhaps the alphabetical order of the cultivars.
Thank you. Yes, we reorganized this and put the same regeneration type from some author together. The date of study order was applied.
The expected advantages of each type of explant should be mentioned.
Discussion/recommendation was added. Like ‘A review of explant sources in rose regeneration studies reveals a preference for leaf-based tissues as the primary material for regeneration studies’
4.4. Media Composition
Lines 251-283 should be ordered according to the component analyzed, for example, salts, carbon source, organic compounds, etc.
Updated based on anther embryo, flower organs, vegetative organs, and root as sequence.
Table 2 does not have the same format as Table 1.
Corrected both tables in the same format.
4.5. Plant Growth Regulators
Table 2 can be mentioned.
Mentioned.
- Organic and inorganic additives
- Case Study
Describe why these cultivars are a model system, what their characteristics are, and why other cultivars are not.
Thank you for your suggestions. We discussed cultivar dependence for regeneration at various parts of manuscript, and these two cultivars are competent rose varieties for regeneration. Requested by another reviewer, we also added ‘Rosaceae crops’ (apple and strawberry) tissue culture regeneration advancements to compare with rose.
Future Prospects and Applications
Lines 402-406 How was this conclusion reached? This topic was not addressed throughout the text, except superficially in the introduction.
The interpretation of the direction of PTC research in roses should be based on all the information compiled by the authors. Which technique has yielded the best results, and why? What is missing? How much progress has been made in addressing what is missing? Where should research be directed? The ideas expressed in this section can be suggested even without reading the text.
We re-organized this part and put focus on rose tissue culture regeneration condition improvement needs to be continued development to meet rose biotech breeding needs. We also propose the application of morphologic regulators in construct, Apical meristem stem bombardment transformation, and floral dip method application in rose transformation may improve rose cell regeneration and genotype-independent biotechnology breeding.
Round 2
Reviewer 2 Report
Comments and Suggestions for Authors
In my opinion, the authors improved the review manuscript in satisfactory form. So, it is accepted in its current format. Thanks
Comments on the Quality of English LanguageThe English could be improved to more clearly express the research.
Reviewer 3 Report
Comments and Suggestions for Authors
The manuscript was substantially improved, despite its initial rejection.
The abstract: placed greater emphasis on the study's rationale, and the study's context was improved in this section.
The introduction: the format used was correct. The information was reorganized, focusing on the study's objective.
Advances and Trends in Rose Tissue Culture (1990–2025): Advances in Regeneration Techniques: relevant information was added.
In most sections, not only were studies listed, but interpretations of these were also included.
Overall, each of the observations made has been addressed, resulting in a substantial improvement to the manuscript. In my opinion, it can be published in its current form.